# Therapeutic Targets of Monoclonal Antibodies Used in the Treatment of Cancer: Current and Emerging

**DOI:** 10.3390/biomedicines11072086

**Published:** 2023-07-24

**Authors:** Brian Effer, Isabela Perez, Daniel Ulloa, Carolyn Mayer, Francisca Muñoz, Diego Bustos, Claudio Rojas, Carlos Manterola, Luis Vergara-Gómez, Camila Dappolonnio, Helga Weber, Pamela Leal

**Affiliations:** 1Center of Excellence in Translational Medicine (CEMT) and Scientific and Technological Bioresource Nucleus (BIOREN), Universidad de La Frontera, Temuco 4811230, Chile; pereznunezisabela@gmail.com (I.P.); d.ulloa09@ufromail.cl (D.U.); c.mayer01@ufromail.cl (C.M.); f.munoz16@ufromail.cl (F.M.); d.bustos06@ufromail.cl (D.B.); lovg002@gmail.com (L.V.-G.); c.dappolonnio01@ufromail.cl (C.D.); hlaweber@gmail.com (H.W.); 2Programa de Doctorado en Ciencias Médicas, Universidad de la Frontera, Temuco 4811230, Chile; crojas.vet@gmail.com (C.R.); carlos.manterola@ufrontera.cl (C.M.); 3Centro de Estudios Morfológicos y Quirúrgicos de La, Universidad de La Frontera, Temuco 4811230, Chile; 4Department of Agricultural Sciences and Natural Resources, Faculty of Agricultural and Forestry Science, Universidad de La Frontera, Temuco 4810296, Chile

**Keywords:** cancer, immunotherapy, monoclonal antibody, therapeutic targets

## Abstract

Cancer is one of the leading global causes of death and disease, and treatment options are constantly evolving. In this sense, the use of monoclonal antibodies (mAbs) in immunotherapy has been considered a fundamental aspect of modern cancer therapy. In order to avoid collateral damage, it is indispensable to identify specific molecular targets or biomarkers of therapy and/or diagnosis (theragnostic) when designing an appropriate immunotherapeutic regimen for any type of cancer. Furthermore, it is important to understand the currently employed mAbs in immunotherapy and their mechanisms of action in combating cancer. To achieve this, a comprehensive understanding of the biology of cancer cell antigens, domains, and functions is necessary, including both those presently utilized and those emerging as potential targets for the design of new mAbs in cancer treatment. This review aims to provide a description of the therapeutic targets utilized in cancer immunotherapy over the past 5 years, as well as emerging targets that hold promise as potential therapeutic options in the application of mAbs for immunotherapy. Additionally, the review explores the mechanisms of actin of the currently employed mAbs in immunotherapy.

## 1. Introduction

Cancer is one of the leading global causes of deceases. In 2020, about 10 million people died due to this condition [1]; by 2040, 29.5% and 16.6% increases are expected in new cases and deaths, respectively [2]. According to the World Health Organization (WHO), cancer is the result of an interaction between genetic factors and three types of external factors: (1) physical carcinogens such as UV and ionizing radiation, (2) chemical carcinogens such as aflatoxins, arsenic, benzopyrene, bisphenol, and tobacco smoke (which are associated with the modern lifestyle), and (3) biological carcinogens such as viruses (e.g., hepatitis B and C, human papillomavirus), bacteria (*Helicobacter pylori*), and parasites (*Schistosoma haematobium*, *Opisthorchis viverrini*, etc.) [3,4]. Recently, lung, colorectum, liver, stomach, and breast cancers (Figure 1) have been reported with the highest mortality rates, representing an alarming public health problem.

Cancer treatment options are constantly evolving and can be grouped into three categories: physical intervention (surgery), radiation therapy and pharmaceutical treatment, which includes chemotherapy, hormone therapy and immunotherapy [5]. The use of monoclonal antibodies (mAbs) in immunotherapy has been considered as the corner-stone of modern cancer therapy [6]. The mAbs have a high capacity to recognize and bind antigens (previously identified therapeutic targets) by its n-terminal extremes and then, coordinate both its inactivation and destruction by its c-terminal effector portion [7]; Moreover, mAbs can be applied alone or linked with drugs, enzymes, radionuclide or others antibodies [8,9,10]. In terms of mechanism of action [11,12], several characteristics make them an attractive therapeutic alternative, including specificity, potency, metabolic stability, malleability and versatility. This alternative enables the specific inhibition of cellular receptors and ligands, either extracellular domains of membrane proteins or secreted proteins [10] (cytokines), which play crucial roles in tumor development and angiogenesis [13]. As a result, out of the currently approved, one hundred and twenty four mAbs, forty-seven are designed for the treatment of cancer [14].

However, to design an appropriate immunotherapeutic regimen for any type of cancer, the identification of specific molecular targets or biomarkers of therapy and/or diagnosis (theragnostic) becomes indispensable in order to avoid collateral damage. Currently, there is an increasing number of established molecular targets for the treatment of diverse types of cancer. This review focuses on the therapeutic targets of mAbs utilized in cancer immunotherapy over the last 5 years and any new emerging targets that are being considered. It is necessary to understand the biological roles of these targets and the function of mAbs in antitumoral immunotherapies and immunotherapeutic treatments.

## 2. Current Immunotherapeutic Targets for Cancer Treatment

Therapeutic targets can be defined as individual molecules or sets of essential molecules involved in the development of one or several pathologies. These targets enable the identification and specific treatment of these diseases. Additionally, therapeutic targets should exhibit differential expression compared to the normal physiological condition, either as soluble ligands or by being expressed on the surface of cells to be accessible to the therapeutic agent [6,15]. This section describes the main therapeutic targets identified in immunotherapy against various types of tumors, which are also represented in Figure 2.

### 2.1. Programmed Cell Death Protein 1 (PD1)/Programmed Cell Death Ligand 1 (PD-L1) Axis

PD1 (or CD279) (UniprotKB—Q15116), is a membrane protein weighing approximately 55 kDa, belonging to the immunoglobulin superfamily. It consists of 288 amino acids (aa) distributed in an extracellular domain (24–170 aa), transmembrane helical domain (171–191 aa), and cytoplasmatic domain (192–288) [16]. This protein is commonly expressed on the surface of cells from immune system, particularly in tumor-specific T cells [17]. When it is activated by its ligands (PD-L1 or PD-L2), PD1 performs its regulatory function by inhibiting T cell-mediated immune responses [18]. This means that activated PD1 recruits the phosphatase SHP2 (UniprotKB—Q06124), which dephosphorylates and attenuates key molecules in the T-cell receptor (TCR) and CD28 pathways. This acts as a checkpoint to control the inhibition of overactive T cells, regulating their activation, killer functions, proliferation, cytokine production, and death [19]. Consequently, our organism is able to control the intensity and duration of T-cell responses in order to maintain self-tolerance and prevent autoimmune attacks [19].

However, several types of cancer cells have developed a strategy of overexpressing the PD1 ligand (PD-L1) on their surface. This allows them to mimic, slow down, and evade the immune response. PD-L1 (or CD274, B7-H1) (UniprotKB—Q9NZQ7) is a 33 kDa membrane protein that also belongs to the immunoglobulin subfamily, and it is the main ligand of PD1. It has 290 aa, which are mainly distributed in two extracellular domains (19–127 aa, Ig-like V-type; 133- 225 aa, Ig-like C2-type), a transmembrane helical portion (239–259 aa), and a small cytoplasmatic domain (260–290) [20]. It is expressed on the surface of regulatory cells such as activated T and B cells, monocytes, keratinocytes, and dendritic cells [21]. It is also expressed by tumoral cells as a strategy to avoid immune response [19]. In fact, PD-L1 expression is observed in different types of human tumor [22], and its involvement in cancer progression has been established. For instance, in kidney cancer, the presence of PD-L1 has been found to induce epithelial-to-mesenchymal transition (EMT) and promote stem-cell-like phenotypes, which are indicative of renal cancer progression [23]. These findings highlight the role of the PD1/PD-L1 pathway in driving the progression of kidney cancer. However, the clinical efficacy of PD1/PD-L1-blocking antibodies has been demonstrated in patients with advanced melanoma, as well as lung and renal cancer. The discovery of the PD1/PD-L1 axis and its role in the control of T cells by Takuso Honjo represents a significant advancement in the actual fight against cancer. For this contribution, the Japanese scientist received the Nobel Prize in Physiology and Medicine in 2018 [24]. Due to these reasons, both PD1 and PD-L1 have been considered valuable therapeutic targets for the use of mAbs against different types of cancer [25,26,27]. Tislelizumab is an mAb that was approved in December 2019 in China for patients with relapsed or refractory classical Hodgkin’s lymphoma after at least second-line chemotherapy [28]. penpulimab, a humanized and engineered mAb (designed to eliminate Fc-mediated effector functions), was approved in August 2021 in China for the treatment of adult patients with relapsed or refractory classic Hodgkin´s lymphoma [29]. However, these two mAbs, along with three others, are currently under review for approval in EEUU and Europe for several types of cancer, such as esophageal squamous cell carcinoma (tislelizumab), metastatic nasopharyngeal carcinoma (penpulimab), non-small-cell lung cancer (sintilimab), nasopharyneal carcinoma (toripalimab), and squamous cell carcinoma of the anal canal (retifanlimab) [14]. Retifanlimab was recently approved by the Food and Drug Administration (FDA) to be used in Merkel cell carcinoma [30]. Dostarlimab (Jemperli) is also one of the most recently mAbs approved (2021) by the FDA and the European Medicines Agency (EMA) for the treatment of endometrial cancer [14]. Furthermore, five other mAbs are currently under review for approval in several types of cancers.

### 2.2. B-Lymphocyte Antigen CD20 (CD20)

CD20 (UniprotKB—P11836) is a 33 kDa membrane protein that belongs to the membrane-spanning 4 domain family A (MS4A). It is expressed on the surface of B cells, although this expression is lost when B cells differentiate in plasmablasts or start secreting antibodies [31]. However, CD20 is found to be expressed in B-cell lymphomas, leukemias, Hodgkin’s disease, and multiple sclerosis, among other conditions [32,33]. Furthermore, it is expressed in approximately 3–5% of CD3^+^ T cells in human peripheral blood [31].

It is composed of N-terminal (1–56 aa) and C-terminal (210–297 aa) domains, along with a small cytosolic domain (106–120 aa), four transmembrane domains (57–78, 85–105, 121–141, and 189–209 aa), and two extracellular domains (79–84 and 142–188 aa). These extracellular domains contain epitopes recognized by most mAbs produced against CD20. However, alternative transcripts encoding truncated forms of CD20 have been identified in malignant B cells, allowing them to evade recognition by mAbs [34,35].

The function of CD20 is not yet fully understood, but is believed to play a crucial role in the optimal development of humoral immunity. It depends on a functional B-cell receptor (BCR) signaling pathway; apparently, CD20 works as a calcium channel that activates BCR [36]. Silencing CD20 in malignant B cells has been shown to affect the phosphorylation of several kinases and proteins associated with BCR [36], suggesting that CD20 is involved in BCR signaling [37].

Due to its prominent presence on the surface of B cells and some T cells, it is considered an important therapeutic target in hematological B-cell malignancies, such as leukemias and lymphomas [35,36], as well as autoimmune diseases, such as multiple sclerosis [32,38]. Two recently approved humanized and bispecific IgG1 mAbs targeting CD20 and CD3 are mosunetuzumab (approved in the European Union) and epcoritamab (approved in the USA), utilized for the treatment of follicular lymphoma and diffuse large B-cell lymphoma, respectively [39,40]. Ublituximab, a chimeric IgG1 mAb against CD20, is currently under regulatory review for approval in the USA, for the treatment of chronic lymphocytic leukemia [14].

### 2.3. Human Epidermal Growth Factor Receptor 2 (Receptor Tyrosine Kinase) (HER2)

The human epidermal growth factor receptor 2 (HER2/ErbB2/Neu) (UniprotKB—P04626), encoded by the *ERRB2* gene, is a glycoprotein with a molecular weight of approximately 185 kDa. It belongs to the ErbB family of transmembrane receptor tyrosine kinases (RTKs), which play an important role in the signaling pathways involved in cell growth, proliferation, migration, differentiation, metabolism, survival, and regulation of intercellular communication during development [41]. It is composed of an extracellular ligand-binging domain (23–652 aa), a hydrophobic transmembrane domain (653–675 aa), and a cytoplasmatic tyrosine kinase domain (676–1255 aa). This kind of RTK is activated when a ligand binds to the extracellular domain, leading to dimerization and autophosphorylation of the cytoplasmatic tyrosine kinase domain. As a result, this initiates downstream signaling, influencing all processes mentioned above [42]. However, HER2 does not have a known ligand; instead, it is constitutively active and able to heterodimerize with other ErbB proteins, thereby becoming a powerful signal transducer [41,42]. HER2 is normally expressed in the epithelia of various organs, and its aberrant overexpression has been associated with adenocarcinomas, including breast, cervix, lung, ovary, endometrium, gastroesophageal junction, gastric, and bladder cancers [43]. Currently, HER2 serves as an important prognostic and therapeutic target for breast cancer. Approximately 15–30% of human breast cancers are HER2-positive or overexpress HER2 [44,45], which is associated with a poorer outcome compared to non-overexpressing cases [41]. To date, two monoclonal antibodies have been approved to treat HER2-positive breast cancer: margetuximab, approved on 2020 by the FDA, and trastuzumab deruxtecan, approved on 2019 in the United States (US) by the FDA and recently (2021) in Europe by the EMA [14].

### 2.4. B-Lymphocyte Antigen CD19 (CD19)

CD19 (UniprotKB—P15391) is a membrane protein with a molecular weight of approximately 95 kDa. It belongs to the immunoglobulin superfamily and is exclusively expressed on B cells [46]. This protein contains 556 aa distributed in an extracellular domain (20–113 aa, Ig-like C2-type1; 176–277 aa, Ig-like C2-type2), transmembrane domain (292–313), and most importantly, a cytoplasmatic domain (314–556 aa). The cytoplasmic domain of CD19 contains conserved tyrosine residues that play an important role in the transduction of CD19-mediated signals [47,48]. It is a critical regulator coreceptor of BCR. Its functions include (a) mobilization of intracellular calcium, which is required for the activation of several transcription factors [49], (b) enhancement of mitogen-activated protein kinase (MAPK) activation, (c) amplification of Src protein tyrosine kinase (PTK) activation, which is involved in the initiation and propagation of BCR signaling, and (d) prolongation of BCR signaling in lipid rafts. For more details of these process, refer to the review by Li et al. [46]. Therefore, CD19 is essential for the primary activation of B cells by T-cell-dependent antigens, as well as for their differentiation into memory B cells [50]. Its abnormal expression, either decreased or not, can result in immune deficiency [46] such as chronic lymphocytic leukemia, follicular lymphoma, and diffuse large B-cell lymphoma, while its increased expression is correlated with systemic sclerosis [51,52]. Tafasitamab (Monjuvi) is an mAb against CD19 approved in 2020 by FDA to treat diffuse large B-cell lymphoma, and it is currently under review by EMA [14].

### 2.5. Disialoganglioside GD2 (GD2)

The GD2 antigen is a sphingolipid coated with five monosaccharides: glucose, galactose, N-acetylgalactosamine, and two N-acetylneuraminic acids [53]. It is synthesized in the Golgi apparatus, and its expression in normal tissues remains restricted to neurons, skin melanocytes, and peripheral pain fibers [54], but it also has been found expressed in stem cells [55]. However, GD2 is overexpressed in several types of cancer such as neuroblastoma [56], small-cell lung cancer [57], melanoma [58], Ewing sarcoma [59], osteosarcoma [60], soft-tissue sarcoma [61], glioma [62], retinoblastoma [63], and breast [64] and bladder cancer [65]. GD2 can be detected in the plasma, peripheral blood, bone marrow, and tumor tissue [53]. Although the general role of GD2 in normal cells is not well defined [66], it is thought that GD2 possesses potent stemness ability [67] because it enhances proliferation, motility, migration, adhesion, and invasion of tumoral cells [68,69,70]. As GD2 overexpression is a signal of malignancy, it is considered an exceptionally promising therapeutic target. Naxitamab (Danyelza) is a humanized mAb against GD2. In 2020, it was approved by the FDA for the treatment of high-risk neuroblastoma and refractory osteomedullary disease [14].

### 2.6. B-Cell Maturation Antigen (BCMA or CD269)

The BCMA (UniprotKB—Q02223) is a membrane protein with a molecular weight of approximately 20.2 kDa, encoded by the *TNFRSF17* gene. It belongs to the tumor necrosis factor receptor superfamily, and it is exclusively expressed on the surface of plasmablasts [71] and plasma cells [72]. This protein contains of 184 aa distributed in an extracellular domain (1–54 aa), transmembrane domain (55–77 aa), and cytoplasmic domain (78–184 aa). This protein plays a crucial role in B-cell proliferation, survival, and differentiation into plasma cells [73]. The extracellular domain of BCMA is cleaved by γ-secretase to produce soluble BCMA, which regulates plasma cell in the bone marrow [74]. The BCMA has two ligands, a B-cell-activating factor (BAFF) and a proliferation-inducing ligand (APRIL). The interaction between APRIL and BCMA transmits differentiation and survival signals, leading to immunoglobulin isotype switching and viability of plasmablasts and plasma cells in the bone marrow [75,76]. BCMA is particularly highly expressed in pathogenic plasma cells from multiple myeloma, B-cell leukemias, and lymphomas [77,78]. It is an ideal target for treating malignancies of these cells types since BCM is elevated in the serum, plasma, and tissues suffering this disease [74,79]. Belantamab mafodotin is a humanized mAb conjugated with a synthetic antineoplastic agent, monomethyl auristatin F [80]. In 2020, it is was approved by the FDA and EMA for the treatment of multiple myeloma [14].

### 2.7. Trophoblast Cell-Surface Antigen 2 (TROP-2)

TROP-2, best known as tumor-associated calcium signal transducer 2, is a cell surface glycoprotein referred to by different names such as epithelial glycoprotein-1, membrane component surface marker-1 (M1S1), cell surface glycoprotein Trop-2, membrane component chromosome 1 surface marker 1, and gastrointestinal antigen 733-1 (GA733-1) [81,82,83] (UniprotKB—P09758). It is encoded by the *TACSTD2* gene, a transmembrane glycoprotein with a molecular weight of approximately 46 kDa. It is composed of an extracellular domain (27–274 aa), small transmembrane domain (275–297 aa), and cytoplasmatic domain (298–323 aa). Due to its ability to be phosphorylated by protein kinase C [84], it is considered an important component of signal transduction across the cell membrane [85]. While its role in normal cells is not yet fully understood [81], TROP-2 has been found to be overexpressed in several solid epithelial cancers, and it is associated with proliferation, cell migration, and tumor growth [85]. The main pathway governing these processes is the PI3K/AKT pathway [86]. In gallbladder cancer, inhibition of TROP-2 was found to regulate the PI3K/AKT pathway by decreasing the expression and phosphorylation of Akt (known to induce oncogenesis [87]) and increasing the expression of the dual phosphatase PTEN [88], an enzyme that suppresses PI3K signaling and AKT activation [87]. Due to these findings, the extracellular domain of TROP-2 has been utilized as an antigen to produce mAbs for therapeutic purposes. Lin et al. isolated a Fab antibody fragment against TROP-2 utilizing phage display technology and successfully inhibited the growth of breast cancer in both in vitro and in vivo experiments [89]. This same Fab antibody fragment was further conjugated with doxorubicin (a commonly utilized in chemotherapy agent), and it was able to inhibit the proliferation and growth of pancreatic cancer in both in vitro and in vivo experiments [90]. Additionally, this Fab fragment was converted into IgG through eukaryotic expression vectors and tested in ovarian cancer, where Liu et al. demonstrated its ability to inhibit tumor cell growth, migration, and invasion in both in vitro and in vivo experiments [91]. Recently, in 2020, sacituzumab govitecan was approved by the FDA for the treatment of triple-negative cancer [14].

### 2.8. Adp-Ribosyl Cyclase/Cyclic Adp-Ribose Hydrolase 1 (CD38)

CD38 encodes for a membrane glycoprotein, with a molecular weight of 34.33 kDa, composed of a cytoplasmatic domain (1–21 aa), transmembrane helix (22–42 aa), and extracellular domain (43–300 aa). This protein is expressed on plasma cells, natural killer cells, several subpopulations of B and T cells, prostatic epithelial cells, pancreatic islet cells, neurons, retinal ganglion cells and another subset of cells [92,93,94,95]. Its main function is the catabolism of nicotinamide adenine dinucleotide (NAD^+^) to cyclic ADP-ribose (cADPR) and cADPR to ADPR [96], as well as the conversion of nicotinamide adenine dinucleotide phosphate (NADP) into nicotinic acid adenine dinucleotide phosphate (NAADP) [97]. Both cADPR and NAADP are considered potent Ca^2+^-releasing messengers implicated in various signaling pathways such as smooth muscle contraction, hormonal secretion, fertilization, immune responses, and other processes [98]. CD38 has been found highly expressed in a subset of hematological tumors, particularly multiple myeloma, compared to low levels on normal cells [99]. Its overexpression leads to a decline in intracellular NAD^+^ and NADP levels, which is believed to disrupt the homeostasis of these important nucleotides, affecting normal metabolic processes and tissue integrity, as well as the tumor microenvironment [100]. Recently, it has been demonstrated that cancer-associated fibroblasts expressing CD38 in melanoma promote disease progression through the production of pro-tumoral factors that enhance tumor cell migration, invasion, and blood vessel formation [101]. In 2020, isatuximab, a chimeric mAb developed by Sanofi that binds CD38, was approved by the EMA and FDA for the treatment of relapsed/refractory multiple myeloma [102,103].

### 2.9. Nectin-4 or Poliovirus Receptor-like 4 (PVRL4)

Nectin-4 (UniprotKB—Q96NY8), also known as PVRL4, is a type of I transmembrane cell adhesion glycoprotein from with a molecular wight of approximately 66 kDa. It is composed of an extracellular domain (32–349 aa), transmembrane domain (350–370 aa), and cytoplasmatic domain (371–510 aa). It belongs to the immunoglobulin superfamily [104]. This protein is involved in the formation and maintenance of cell-to-cell adhesion junctions, and it participates in both homophilic and heterophilic interactions with cadherins. It regulates processes such as polarization, cellular adhesion, and movement [105,106]. In regular human tissue, Nectin-4 is predominantly expressed in the placenta and embryo [104,107,108]. However, under pathological conditions, it can be expressed in the skin, esophagus, stomach, bladder, breast, salivary gland, trachea, prostate, and lung [106,109]. Nectin-4 has been shown to be overexpressed in several types of cancer, leading to the activation of WNT-β catenin and Rac small protein pathways in the PI3K/AKT pathway [110]. While Nectin 4 mRNA expression is absent in healthy tissues, its presence in triple-negative breast cancer (TNBC) has been associated with a poor prognosis [111]. The serum levels of Nectin-4 in lung cancer patients and the expression in urothelial carcinoma cells are significantly higher compare to healthy patients and cells, respectively [112]. However, the interaction of Nectin-4 with tumor microenvironments and its predictive and prognostic role are still controversial [112,113]. Despite this and due to its overexpression, Nectin-4 was utilized to create enfortumab vedotin, a potent antibody conjugated to an antimitotic agent, denominated monomethyl auristatin E (MMAE). This conjugate disrupts microtubules and induces apoptosis in multiple preclinical cancer models [109], particularly in urothelial carcinoma cells [112].

### 2.10. Cd79b (ADC) Diffuse Large B-Cell Lymphoma

CD79b (UniprotKB—P40259), also known as IgB, is a glycoprotein with a molecular weight of 26 kDa, encoded by the B29 gene [114]. It is part of the heterodimeric signaling component of the BCR [115], and it is exclusively expressed by the B-cell compartment. It is composed of an extracellular domine (29–159 aa, Ig-like V-type), transmembrane helix (160–180 aa), and small cytoplasmatic domain (181–229 aa). The cytoplasmatic region contains an immunoreceptor tyrosine-based activation motif (ITAM) [116], which is responsible for initiating BCR aggregation [114]. CD79b is responsible for mediating the surface expression and signaling of various BCR complexes at all stages of development, including immature precursor B cells and mature B cells [117,118]. Mutations in this protein have been associated with different types of bone marrow cancer. CD79B mutations in conjunction with MYD88L265P occur in approximately 8% of diffuse large B-cell lymphomas [119]. Moreover, the presence of the CD79b protein is commonly observed in patients with chronic lymphatic leukemia and in some cases of multiple myeloma. It also serves as a reliable pan-B-cell marker for the detection of neoplastic B-cells [120]. CD79b is expressed in non-Hodgkin’s lymphomas about 90% of the time [121,122]. The ITAM domain enables the initiation of signaling cascades, leading the translocation of NF-κB members to the nucleus and subsequent transcription of pro-survival target genes. This promotes cell survival and proliferation, sustaining neoplastic proliferation [123]. Polatuzumab vedotin, a humanized mAb covalently conjugated with MMAE via a cleavable linker was developed by Genentech (a subsidiary of Roche) [124]. It was designed to bind CD79b. In 2019 and 2020, it was approved by the EMA and FDA, respectively, for the treatment of diffuse large B-cell lymphoma [14].

### 2.11. CD22

CD22 (UniprotKB—P20273) is also known as B-lymphocyte cell adhesion molecule (BL-CAM), sialic acid-binding Ig-like lectin 2 (Siglec-2), and T-cell surface antigen Leu-14. It is a 155 kDa protein that belongs to the SIGLEC family of lectins. It is a single-pass type I membrane protein, with an extracellular domain (20 to 687 aa), single transmembrane domain (688 to 706 aa), and cytoplasmic domain (707 to 847 aa). It is expressed during the early stages of ontogenesis of B cells in the spleen and bone marrow [125]. It is known to be an inhibitory receptor [126,127], and it is present exclusively on B cells, predominantly on mature B cells, where it regulates proliferation and function [128]. By associating with the BCR, CD22 inhibits B-cell response, preventing an overly aggressive B-cell reaction and autoimmunity [126]. One of its functions is to prevent the development of autoimmune diseases, although altered B cells can bypass the normal checkpoints and facilitate the development of autoimmune diseases [129], such as systemic sclerosis [130] and systemic lupus erythematosus [131]. By binding to sialo glycans, it reduces the regular inhibitory effects [126]. It is also known to promote cell proliferation and apoptosis via BCR [128]. It is often found alongside CD19 and CD20 on the surface of hyperactivated B cells in autoimmune diseases. Consequently, Medimmune (a parent company of AstraZeneca) has developed some therapeutic approaches, including naked antibodies [132], antibody–drug conjugates [133], radioimmunoconjugate antibodies [134], bispecific antibodies [135], and trispecific antibodies [136]. Moxetumomab pasudotox is a murine mAb conjugated with a toxic fragment of Pseudomonas exotoxin A that specifically targets CD22. It was approved by the FDA in the USA in 2018 and received approval by the EU in 2021 for the treatment of hairy cell leukemia [14,137].

### 2.12. CC Chemokine Receptor Type 4 (CCR4)

CCR4 (UniprotKB—P51679) is a chemokine receptor that can recognize CCL17 (thymus- and activation-regulated chemokine) and CCL22 (macrophage-derived chemokine) [138]. This antigen is composed of 360 aa distributed in four extracellular domains (1–39 aa, 99–111 aa, 176–206 aa, and 268–284 aa), seven transmembrane domains (40–67 aa, 78–98 aa, 112–133 aa, 151–175 aa, 207–226 aa, 243–267 aa, and 285–308 aa), and four cytoplasmatic domains (68–77 aa, 134–150 aa, 227–242 aa, and 309–360 aa). CCR4 is primarily expressed by Th2 lymphocytes and regulator T cells (Treg), and it is often overexpressed in mature T-cell cancers such as adult T-cell leukemia (ATL) and cutaneous T-cell lymphomas (CTCLs) [139]. In tumors, blocking of CCR4-dependent Treg recruitment is an important mechanism of tumor-extrinsic immune resistance [140]. Furthermore, accumulation of Treg cells in the tumoral microenvironment (TME) weakens the immune response of patients receiving immunotherapy [141]. The degree of infiltration of CCR4^+^ Treg cells in prostate cancer is related to prognosis, with higher levels of expression of CCR4^+^ T cells observed in specimens with higher Gleason scores (≥8) (*p* = 0.041) and associated with a faster progression to castration-resistant prostate cancer [142]. CCR4 has been identified as a tumor-initiating chemokine in cutaneous T-cell lymphoma [143], associated with migration and invasion of lung cancer cells [144], promoting metastasis (together with CCL17) in bladder cancer [145] and facilitating cell migration (together with CCL2) in head and neck squamous cell carcinoma [146]. The full therapeutic potential of CCR4 inhibition and Treg depletion is still to be defined [147]. However, mogamulizumab, a humanized anti-CCR4 mAb developed in Japan, was approved in 2018 by the FDA and EMA for the treatment of cutaneous T-cell lymphoma. This mAb has been used in combination with durvalumab (anti PD-L1) or tremelimumab (anti CTLA-4) for the treatment of solid tumors [138]. It has also been labeled with indium-111 for in vivo diagnostic imaging [148].

### 2.13. PDGRFα

PDGRFα (platelet-derived growth factor subunit A) (UniprotKB—P04085) belongs to the type III transmembrane receptor tyrosine kinase (RTK) family, and it is a component of platelet-derived growth factor receptors (PDGFRs) [149]. Its structure is characterized by five extracellular domains named D1–D5, which are similar to immunoglobulins. It is also composed of a transmembrane portion and two intracellular tyrosine kinase domains with an adenosine triphosphate (ATP)-binding region and a phosphotransferase region [150,151]. Upon binding with its ligand, PDGFRα undergoes receptor dimerization, leading to autophosphorylation of the tyrosine kinase domain and subsequent activation of pathways involved in cell-cycle activation, cell proliferation, and apoptosis inhibition, such as PI3K/Akt, Ras/MAPK, JAK/STAT, and RaF/MEK/ERK [150,152,153]. This receptor plays a crucial role in the regulation of biological processes, including embryonic development, gastrulation, and organ development such as the lungs, intestine, skin, testicles, and kidneys [154,155]. It is responsible for the maintenance of mesenchymal stromal cell (MSC) and immune infiltration [156,157,158]. Moreover, it is involved in cell proliferation, migration, invasion, and tumor progression [155,156,159]. Errors in the activation of PDGFRα associated with malignant tumors by mutation, amplification, and gene fusion have been observed. Mutations commonly occur in gastrointestinal stromal tumor cells (5–10%) [160], as well as in non-small-cell lung cancer (6%) and colorectal cancer (5%). Gene amplification problems are frequently seen in glioblastoma (12%) [161,162]. Increased expression of PDGFRα has been shown to stimulate proliferation, metastasis, and invasive potential in papillary thyroid cancer cells. In addition, this high expression is associated with a lower probability of patient survival, highlighting its potential as a biomarker and pharmacological target in thyroid cancer therapy [163,164].

### 2.14. SLAMF7 (CD319)

SLAM (signaling lymphocytic activation molecule) is a family of receptors expressed exclusively at varying frequencies on immune cells, including CD8 T cells, natural killer cells, and activated B cells [165]. One member of this family is SLAMF7 (SLAM family member 7), which is also known as CD319, CRACC, CS1, and 19A. It is a protein encoded by the SLAMF7 gene (UniprotKB—Q9NQ25). Except for SLAMF2, SLAMF proteins are type I transmembrane glycoproteins [166]. They are composed of an extracellular domain (23–226 aa), helical transmembrane domain (227–247 aa), and cytoplasmic domain (248–335 aa). There are seven isoforms of SLAMF7 produced by alternative splicing which differ in the extracellular (isoforms 2, 4, and 7), transmembrane (isoform 4, 6, and 7), and intracellular (isoforms 3, 5, and 7) domains. SLAMF7 functions as homotypic receptor, binding SLAM-associated proteins (e.g., EAT-2) through its cytoplasmic immunoreceptor tyrosine-based switch motifs [167]. This interaction leads to the activation of cellular immune responses [168], including natural killer cell activation [169]. However, SLAMF7 can also be activated in the absence of EAT-2, resulting in cellular inhibition [170]. In this context, SLAMF7 has been implicated as a regulator of T-cell inhibitory programs and a contributor to T-cell exhaustion [171]. Overexpression of SLAMF7 in clear-cell renal cell carcinoma (ccRCC) has been associated with poor patient survival due to T-cell exhaustion [171]. More than 95% of multiple myeloma patients express SLAMF7 [172], making it a promising therapeutic target in multiple myeloma. Elotuzumab, a humanized mAb, was approved in the USA in 2015 and EU in 2016 for the treatment of multiple myeloma [14].

### 2.15. Tissue Factor/CD142

Tissue factor/CD142 (TF/CD142, UniprotKB—P13726), also known as coagulation factor III and thromboplastin, is a protein with a molecular weight of 33–27 kDa, encoded by the F3 gene in humans. There are two reported isoforms, P13726-1 and P13726-2. Isoform 1, also known as flTF, is a 33.07 kDa protein located on the cellular membrane. It is composed of an extracellular domain (33–251 aa), transmembrane domain (252–274 aa), and cytoplasmic domain (275–295 aa). Isoform 2 (UniprotKB—P13726-2), also known as asHTF, is a 27.145 kDa protein secreted from the cell. The amino-acid sequence of isoform 2 varies from the canonical sequence (isoform 1) from positions 199 to 238, and it is missing from positions 239 to 295. Its main function is to initiate normal blood coagulation by forming a complex with circulating factors VII or VIIa [173]. Additionally, it promotes maturation, cell growth, development, differentiation, migration, and cell mobility [174] by stimulating the production of chemokines such as interleukin-8 [175], CCL2 (C–C motif ligand 2) [176], and KC (keratinocyte-derived chemokine) [177]. CD142 has been reported to be expressed on the surface of multiple cancer cells, and it is also associated with the mobility of several tumors, including gastric adenocarcinoma [175], colorectal cancer [174], and ovarian cancer [178]. Although it has low tissue specificity, it has been identified as a marker in gastric adenocarcinoma [175], colorectal cancer [174], and prostate cancer [179]. It is also known to promote angiogenesis, endothelial cell proliferation, and other processes. The expression of CD142 is dependent on cell type and can be induced by interleukin-1 [180], interleukin-33 [181], and TNFα [182]. In 2021, the FDA approved an mAb denominated tisotumab vedotin (Tivdak^TM^). It is a human mAb conjugated with MMAE that binds tissue factors expressed in metastatic cervical cancer [183].

### 2.16. CTLA-4

CTLA-4 (UniprotKB—P16410) is a membrane protein with a molecular weight of approximately 25 kDa, associated with T cytotoxic lymphocytes [184]. It is considered an immune checkpoint and an important negative regulator of T-cell responses [185]. It belongs to the immunoglobulin superfamily and contains 223 aa distributed in an extracellular domain (36–161 aa), transmembrane helical domain (162–182 aa), and cytoplasmatic domain (183–223 aa). The CTLA-4 pathway is involved in regulating the immune response mediated by T cells during the priming phase [185]. CTLA-4 is constitutively expressed on regulatory T cells to exert immune suppression [185,186]. However, in the cell, 90% of CTLA-4 in early stages as inactive T-cells is located in the stimulating signals of CD28. The translocation of CTLA-4 is induced through exocytosis [186,187], thus regulating the level of expression of CD28 [188].

CTLA-4 plays an important role in tumorigenesis and tumor immunity. Therefore, it serves as a prognostic biomarker in different types of cancer [184,189]. Liu et al. observed increased levels of CTLA-4 in 13 tumor tissue types, including endometrial carcinoma cervix, cholangiocarcinoma, invasive breast carcinoma, head and neck carcinoma, esophageal carcinoma, renal papillary cell carcinoma, clear-cell carcinoma of the kidney, adenocarcinoma of the lung, hepatocellular carcinoma of the liver, adenocarcinoma of the liver, adenocarcinoma of the colon, squamous cell carcinoma of the lung, and adenocarcinoma of the prostate [184]. The levels of CTLA-4 were correlated with the degree of infiltration of cells such as T and B cells, macrophages, dendritic cells, and neutrophils in different types of cancer [184].

Usually, CTLA-4 therapy is combined with the marker PD-1, resulting in improved survival of patients with liver cancer [185]. However, the response rate to treatment in patients remains low (<50%) [190]. Currently, there are two anti-CTLA-4 human mAbs. The first is ipilimumab, developed by Bristol-Meyers Squibb Pharmaceuticals (New York, NY, USA) and approved in the USA and EU, in 2011, for the treatment of metastatic melanoma [191]; it is used in gastro-esophageal cancer [192], colorectal cancer [193], and lung cancer [194]. The second is tremelimumab, developed by Pfizer and approved in the USA, in 2022, for the treatment of liver cancer and others cancers [195].

## 3. Emerging Immunotherapeutic Targets for Cancer Treatment

Thanks to advances in proteomics and the analysis of extracellular vesicles secreted in the blood plasma of patients with different types of cancer, the identification of potential protein biomarkers for immunotherapy treatment has become faster and more efficient. Some emerging proteins that play a crucial role in the progression and malignancy of various types of cancer are described below, which are also being considered for the design of mAbs for immunotherapy treatment.

### 3.1. COL11A1

Alpha-1 collagen (XI) (UniprotKB—P12107) is a polypeptide chain encoded by the COL11a1 gene. It is composed of 1806 aa, and it is one of the alpha chains that make up alpha collagen XI [196], which is a heterotrimer composed of alpha chains encoded by the COL11a1, COL11a2, and COL2a1 genes [197]. This protein belongs to the cartilage family [198], and it is classified as a minor fibrillar collagen subgroup. It is composed of various domains but does not form triple-helical domains [199]. It has a globular amino-acid domain called TSPN (32–299 aa) [200] and a C-terminal propeptide, known as the COLFI domain (1575–1804 aa), which serves as a binding site for different proteins via calcium ions (UniprotKB, 2021).

COL11a1 is a critical protein involved in the regular formation of collagen fibrils and the regulation of type II collagen fibrillogenic in different mammalian models [201]. It is predominantly expressed in the extracellular matrix [198], and it can be found in different tissues, including the articular cartilage, testicles, trachea, tendons, trabecular bone, skeletal muscle, placenta, and lung [201]. However, its expression in these tissues is relatively low. Interestingly, overexpression of COL11a1 is associated with different types of aggressive cancers, resistance to chemotherapy [198], and low prognosis [197]. This is particularly evident in mesenchymal tumors derived from scleroderma and keloids, as well as in gliomas/glioblastomas in humans, which exhibit high levels of COL11a1 expression [202].

Among the types of related cancers, lung cancer [203] is noteworthy, as COL11a1 has been associated with metastasis to lymph nodes and a poor prognosis. It promotes metastasis and resistance to cisplatin [204]. Similarly, overexpression of COL11a1 has been related to a poor prognosis in ovarian cancer [205]. This is related to increased level of metastasis and resistance to chemotherapy, primarily observed in stromal cells and, in particular, fibroblasts associated with cancers [206].

Furthermore, COL11a1, along with other proteins present in the extracellular matrix, is overexpressed in breast carcinoma. These proteins are released into the blood, enabling their detection in plasma using the ELISA technique [204,206]. Upon review, COL11a1 can be highlighted as a potential biomarker for different types of cancers. Its overexpression is associated with more aggressive cancers, poor prognosis and resistance to chemotherapy. However, there currently no specific alternatives designed to inhibit its function.

### 3.2. Claundin 18

Claudin 18 (UniprotKB—P56856), also called CLDN18, is a membrane protein that belongs to the claudin family. It is composed of 261 aa distributed across two extracellular domains (28–80 aa and 144–174 aa), four transmembrane domains (7–27 aa, 81–101 aa, 123–143 aa, and 175–195 aa), and three intracellular domains (1–6 aa and 102–122 aa). Its main function is associated with the maintenance of tight junctions, which regulate the exchange of molecules between cells [207]. Claudins are predominantly found in gastric [208], pancreatic [209], and pulmonary [210] tissues. Claudin 18 has two isoforms: CLDN18.1, mainly expressed in the lung; CLDN18.2, a specific isoform overexpressed in the stomach, which has emerged as an ideal biomarker [207] because it is widely expressed only in cancer cells, particularly gastric and gallbladder cancer [15]. CLDN18.2 is retained in the presence of a malignant transformation, making it an ideal candidate for monoclonal antibody binding [211]. Changes in claudins at tight junctions are associated with damage to tight adhesions and polarity in the epithelium [212]. These structural abnormalities can lead to increased cell proliferation, epithelial–mesenchymal transition, invasion, and metastasis [213]. Furthermore, the expression of CLDN18 is correlated with a common malignant Epstein–Barr virus-associated tumor known as EBV infection, specifically in gastric cancer (EBVaGC) [214]. This correlation is supported by the fact that the majority of EBVaGC cases exhibit high levels of CLDN18.2 expression. The CLDN18.2 expression in tumor cells is likely associated with key features of EBV12-mediated carcinogenesis [215]. On the contrary, low expression of CLDN18.2 is associated with changes in mucin expression, which is used to classify GC into different mucin phenotypes [216]. CLDN18.2 is not expressed in any healthy tissue, except for the gastric mucosa [216]. Recently, zolbetuximab, a chimeric IgG1 mAb, which binds to CLDN18.2 on the surface of tumor cells, was developed and investigated in clinical trials [217,218]. This mAb triggers antibody-dependent cellular cytotoxicity (ADCC) and induces apoptosis and inhibition of cell proliferation [207]. As a first-line treatment, it showed improved median survival in patients with CG expressing claudin 18.2 compared to chemotherapy alone [219,220]. For this reason, CLDN18.2 is being considered as a potential new target in several types of tumors, given the remarkable success of zolbetuximab against GC.

### 3.3. CD73

CD73, also known as ecto-5’-nucleotidase, is a protein composed of 574 aa and encoded by the NT5E gene (UniprotKB—P21589). It has a molecular weight of approximately 63.4 kDa and is found in the cell membrane with hydrolase activity. It is anchored to glycosylphosphatidylinositol (GPI). It is expressed at different levels in tissues, as well as cells such as endothelial cells, epithelial cells, and T and B lymphocytes. In these cases, it acts as an important factor in the differentiation of these two lines [221,222]. A soluble form of CD73 has been reported, which is responsible for transforming extracellular ATP into immunosuppressive adenosine. This process is correlated to CD39 and limits immune activity, leading to a rare disease denominated “arterial calcifications due to CD7 deficiency” [222].

Several studies have reported that CD73 is upregulated in various types of cancer, and that a higher level of CD73 is commonly associated with worse clinical outcomes [223]. When CD73 is overexpressed in cancer cells, they begin to generate high levels of adenosine, creating a microenvironment in the tumor area [224]. This adenosine-rich microenvironment promotes the growth and proliferation of cancer cells, angiogenesis, and immune suppression in the area [224]. Consequently, CD73 acts as an immunoinhibitory protein that promotes tumor metastasis [224,225]. Due to this role, CD73 is considered an important target for inhibiting tumor growth in many therapies.

CD73 can affect different tumorigenic characteristics, such as cell proliferation, by regulating the cell cycle. It also plays a role in apoptosis and other signaling pathways, including EGFR, beta-catenin/cyclin D1, VEGF, and AKT/ERK [225], via the Rap1/P110β pathway [226]. Additionally, it is also responsible for processes such as adhesion, migration, stemness, angiogenesis, and metastasis [225]. It exhibits enzymatic activity related to CD39, as it utilizes AMP as a raw material for metabolic processes, resulting in the production of adenosine, which acts on extracellular receptors, regulates the activity of adenylate cyclase (AC), and can be integrated into the cell through nucleoside transporters [225].

Adenosine functions as an immunomodulatory factor generated by the degradation of ATP by NTPDase1 [224]. While ATP mediates inflammatory responses, adenosine acts as an anti-inflammatory mediator, regulating the decline of immune cell function through its four receptors (A1, A2A, A2B, and A3) [222,224].

Due to its critical role in antitumor immunity, CD73 has been identified as a promising target for immunotherapies. Recent research has established effective models using strategies such as anti-CD73 mAb or inhibitors such as APCP, LY3475070, AB680, and CB- 708. These approaches have demonstrated the antitumor effects of CD73 inhibition in preclinical trials tested in mice [227,228]. CD73 has also been found to contribute to chemoresistance against doxorubicin carboplatin, gemcitabine, and paclitaxel [229].

### 3.4. B7-H3 (CD276)

B7-H3 (UniprotKB—Q5ZPR3) antigen, also known as B7 homolog 3, is encoded by the CD276 gene, with a molecular weight of 45–66 kDa [230]. It is a type I transmembrane protein that belongs to the B7 ligand family, sharing 30% aa identity with PD-L1. Similar to PD-L1, it is also considered an immune-checkpoint protein [231].

B7-H3 is composed of an extracellular region (29 to 466 aa), short transmembrane domain (467 to 487 aa), and final cytoplasmic domain (488 to 534). It has two isoforms: 2IgB7-H3 (In soluble form) and 4IgB7-H3, which are caused by exon duplication [232].

Its primary biological function is the interaction with the CD28 receptor on T cells in order to co-stimulate deregulation, differentiation, and activation of T cells [232]. Normally, B7-H3 is expressed at low levels in fibroblasts, progenitor cells, and immune cells [233] to regulate the immune system and maintain self-tolerance.

Conversely, B7-H3 has been found to be overexpressed in several types of cancer, including ovarian [234], cervical [235], colorectal [236], breast, lung, brain [237], and neuroblastoma [238], associated with promoting tumor growth, metastasis, drug resistance, and poor survival rates [239,240,241].

The Memorial Sloan Kettering Cancer Center (MSKCC) has developed a murine monoclonal antibody that binds B7-H3 in neuroblastoma cells. Although it is still under review by the FDA an EMA, the FDA has granted it breakthrough therapy designation due to the lack of approved drugs for the treatment for neuroblastoma or targeting this emerging antigen [242].

### 3.5. Interleukin-13 (IL13)

IL-13 (UniprotKb—P35225) is a cytokine belonging to the group of lymphokines. It has a length of 146 aa and a molecular weight of 15.8 kDa. It is a structural cytokine highly related to IL-4, and both participate in immune regulation; they have also been found in pregnancy, fetal development, breast development in infancy, and important higher brain functions such as learning and memory [243].

Both interleukins regulate various cellular functions and activate the transcriptional machinery through cell surface receptors [244]. They are essential for the induction and persistence of the type 2 immune response, and they are associated with multiple atopic diseases [245]. Overexpression of IL-13 and its receptor IL-13R is correlated with the pathogenesis and progression of various malignancies [246]. These interleukins are primarily produced by immune cells, such as CD4 T cells, TH2 cells, basophils, eosinophils, and NKT cells [244].

IL-13 has three different types of receptors and shares two receptor chains with IL-4, allowing them to regulate both common and diverse biological functions [247]. The IL-13 receptors include the primary IL-13R (IL-13Rα1/IL-13Rα2) and the secondary receptor (IL-4Rα/IL-13Rα1), which are expressed in nonhematopoietic cells [247]. The type III receptor (IL-4Rα/IL-13Rα1/ γc) is exclusively expressed on the surface of hemocytes, providing a wide range of complex signaling pathways regulated by IL-13 and IL-4 [247].

In the case of the alpha 2 receptor chain (IL-13Rα2), it interacts not only with interleukin-13 but also with other proteins, such as chitinase-3-like protein 1. However, it has high affinity for IL-13 [248].

IL-13R receptors have been found to be produced in various types of cancer, and their appearance is a result of altered cytokine signaling pathways, induced by the inflammation produced by cancer cells [249]. Furthermore, it has been observed that IL-13R receptors are overexpressed in several cell lines of solid human cancer, including pancreatic cancer and ovarian cancer [250].

IL-13 Rα2, one of the most commonly observed IL-13 receptors in cancer-related cell lines, has been directly related to metastasis in GC. Its overexpression is associated with increased invasion of tumor cells into other tissues [251,252], and it is considered a poor prognosis factor [253]. Another IL-13 receptor that has been observed is IL-13Rα1, which has been found in patients with colorectal and gallbladder cancer lesions [248].

A recent therapy targeting the overexpression of IL13R is the development of IL13 immunotoxins combined with highly cytotoxic truncated proteins. The aim is to reduce the amount of IL13R present in cancers [248], despite this form of therapy being highly cytotoxic to surrounding cells [248]. The most promising approach is the use of specific monoclonal antibodies against the IL-13R receptor chain to reduce its presence in cancerous tissues, thereby reducing metastasis and cell proliferation [248,250].

## 4. Mechanisms of Action of the mAbs Currently Used in the Treatment of Cancer

The functional effect of an mAb in cancer treatment is dependent on the cancer antigen profile and the specific mechanisms of action of mAb. These mechanisms can include blocking the ligand or receptor, internalization of the mAb, activation of Fcγ receptors (FCGR) on innate immune cells, activation of complement, or blocking receptor-mediated oncogenic signaling. This section provides a brief description of some of these processes, as observed in antibodies currently approved for the treatment of cancer.

### 4.1. Blocking Ligand Binding

These mAbs bind to ligands or receptors on the cellular surface, preventing the ligand from binding to the receptor [254]. However, the mechanism can vary depending on whether the antibody is conjugated or not.

#### 4.1.1. Nonconjugated mAbs

Antibodies that are not conjugated can exert their function through several mechanisms:Steric hindrance: the antibody binds to the antigen receptor or ligand, occupying the region of interaction. By physically blocking the binding, it prevents the transmission of signals or the initiation of unwanted biological responses [255].Conformational changes: the antibody binds the antigen and induces conformational changes in the target cell. This alteration in structure prevents efficient interaction with ligand [255].Internalization of the complex: in some cases, once the antibody has bound to the antigen, the antibody–antigen complex undergoes internationalization. The cell captures the complex through endocytosis, removing the receptor and ligand from contact with each other. This blocks the signaling or biological activity mediated by them [256].

#### 4.1.2. Conjugated mAbs

Conjugated mAbs are a class of therapies that combine the selective benefits of mAbs with the ability to transport and release specific therapeutic agents. These therapeutic agents can include cytotoxic molecules, radioisotopes, drugs, cytotoxic payloads, or even other therapeutic proteins [256,257]. The working mechanism of mAbs conjugated involves the following steps: selective mating, internalization, and release of the therapeutic agent.

Once the conjugated mAb has bound to the specific antigen or receptor on the surface of the target cell, it can be internalized by the cell through endocytosis. Once inside the cell, the antibody targets specific intracellular compartments, such as endosomes or lysosomes [258,259]. Within these compartments, the therapeutic agent bound to mAb is released. It can occur through various mechanisms, such as enzymatic degradation, pH changes, or the action of proteases. The release of the therapeutic agent enables its specific activity within the target cell [257,259].

##### Action of the Therapeutic Agent

Once released within the cell, the therapeutic agent can exert its specific effect. For instance, if the therapeutic agent is a cytotoxic molecule, it can induce programmed cell death (apoptosis) or interfere with vital cellular processes required for cell survival. On the other hand, if the therapeutic agent is a drug, it can obstruct specific signaling pathways or disrupt metabolic processes necessary for cell proliferation or survival [258].

### 4.2. Blocking Signaling Pathway

Furthermore, mAbs can induce the death of tumor cells by blocking the signaling pathways associated with growth factor receptors. This can be achieved when the Fab region of the mAb recognizes the receptors for the growth factors, resulting in the inactivation of signaling pathways or blocking the binding of the ligand [260]. The mAb impedes tumor cell survival cascades, interfering with cell proliferation, adhesion, and angiogenesis, eluding programmed cell death, and evading immune checkpoints [261,262]. Tumor signaling can be perturbed when targeted antibodies disrupt growth signaling pathways by neutralizing cytokines that are critical to cellular growth and proliferation [263].

While the mechanisms of action (MOAs) for these types of mAbs are similar, the subsequent effects vary due to a range of intrinsic properties. This properties include the mAb-binding epitope, affinity, and serum half-life [264]. The mAb can alter dimerization properties, resulting in different signaling properties depending on whether it is targeting a homodimer or a heterodimer receptor. Understanding this complexity is crucial as it has a significant impact on the development and clinical testing of novel therapeutics, particularly those involving mAb combinations [265].

The majority of FDA-approved mAbs target two members of the ERBB family, HER2 and EGFR. Both EGFR and HER2 are cell surface membrane-spanning Type I receptors, highly expressed on different solid cancers and capable of triggering a wide range of oncogenic signaling through homo- and heterodimerization [259,261,266]. Although HER2-targeted mAbs were initially described as inhibitors of HER2-mediated signaling, multiple studies have demonstrated that these mAbs also inhibit the downstream PI3K/Akt signaling pathway, resulting in p27 upregulation and inhibition of cellular proliferation [259,267]. Subsequent studies have revealed that HER2 mAbs primarily exert immunologic MOAs by engaging Fc receptors to activate innate immune effector functions and complement activity [259,268].

Trastuzumab and pertuzumab are examples of HER2 mAbs. Trastuzumab recognizes domain 4 of HER2, while pertuzumab binds to domain 2. Pertuzumab specifically prevents both hetero- and homodimerization of HER2 with EGFR. This blocking of receptor tyrosine kinase dimerization results in the shutdown of signaling, leading to the inhibition of cell proliferation, which is a consequence of activated signaling [264].

EGFR is a transmembrane glycoprotein. It is composed of an extracellular ligand-binding domain and a cytoplasmic domain housing a tyrosine kinase. Cetuximab is the most studied anti-EGFR agent, which exerts its effects by blocking ligand binding and receptor dimerization, leading to cell-cycle arrest and apoptosis in tumor cells [263]. On the other hand, panitumumab acts as an antagonist and induces the internalization of EGFR. By preventing intracellular processes triggered by EGFR activation (e.g., dimerization, autophosphorylation, and signal transduction), panitumumab promotes an increase in the apoptotic rate and reduces the proliferation and angiogenesis of tumor cells [260,269].

The mAb targets inhibitory immunologic checkpoint signals, thereby enhancing the antitumoral cellular immune response [265]. Therapeutic mAbs that target coinhibitory receptor pathways (e.g., CTLA-4 or PD-1/PD-L1) have been shown to limit T-cell exhaustion, enhance CD8^+^ T cell antitumor activity, and increase the ratio of effector T cells (Teff) to regulatory T cells (Treg) within tumors [270].

### 4.3. Depletion of Target by Fc Interaction

As we know, mAbs can perform their function in multiple ways, particularly in the treatment of cancer. One of these mechanisms involves the interaction between the Fc domain of immunoglobulins and Fc receptors (FcRs) present on effector cells [271], such as natural killer (NK) cells, cytotoxic T cells, dendritic cells, monocytes, neutrophils, and macrophages. Through this interaction, mAbs can mediate antibody-dependent cellular cytotoxicity (ADCC), antibody-dependent phagocytosis (ADP), and complement-dependent cytotoxicity (CDC) [258,272].

ADCC occurs mainly when the antibody binds to FcRs on NK cells. It initiates multiple signals that lead to release of lytic compounds from the effector cell, resulting in lysis of the target cell [271]. ADP take place when antibodies opsonize a cell, facilitating its recognition and internalization by phagocytic cells such as macrophages, neutrophils, dendritic cells, and monocytes. Within the phagolysosomes of these cells, the opsonized pathogens or malignant cells are internalized and degraded [271]. CDC begins when the immunoglobulin binds to the antigen on the target cell.

Subsequently, the Fc region of the antibody recruits protein such as C1q, C1r, C1s, and serine proteases, initiating a proteolytic cascade that generates cell death [273,274].

For this reason, immunoglobulins IgG are the most commonly used for treatment of cancer. There are four subtypes of IgG: IgG_1_, which is the most abundant in the plasma and can induce strong ADCC, ADP, and CDC [275]; IgG_2_, which self-aggregates, becoming less useful when linked to drugs [276]; IgG_3_, which is not widely used due to its shorter half-life (around 7 days) compared to the other subtypes with a half-life of 21 days [277]; IgG_4_, which is used in some cancer treatments due to its unique dynamism [278].

In Table 1, the antibodies used in the treatment of cancer in the last 5 years are listed and described.

## 5. Perspectives

It is evident that the emergence of immunotherapy has significantly improved the prognosis and life expectancy of patients with various diseases, particularly cancers. Immunotherapeutic strategies encompass cancer vaccines, oncolytic viruses, adoptive transfer of ex vivo activated T and natural killer cells, and administration of recombinant proteins and primarily mAbs that either co-stimulate cells or block checkpoint pathways [310]. Despite the benefits of mAbs in cancer treatment, accessibility remains an issue in underdeveloped countries and even in developed ones due to the high cost.

For instance, the humanized mAb dostarlimab (Jemperli), which targets the PD-1 receptor, shows promise in treating colon cancer, eradicating all tumor cells within 6 months of treatment in a small yet promising clinical trial involving colorectal cancer patients. The treatment involves one dose every 3 weeks for 6 months [311]. However, each dose costs just over 11,000 USD, making the total treatment cost around 88,000 USD. Consequently, strategies that enable more affordable production of therapeutic antibodies are crucial. The widespread implementation of directed evolution techniques, such as phage display, can help reduce production costs and facilitate the production of fully human antibodies. However, to effectively apply this technology, it is essential to identify and validate the different overexpressed antigens as cancer biomarkers, determining their biological functions, topology, and the extracellular regions or the most exposed areas susceptible to antibody recognition, as proposed in this study. Subsequently, a panel of these extracellular regions or areas that are prone to antibody recognition can be recombinantly produced. These antigens can be used for the selection and isolation of full human variable domains of mAb from recombinant naïve [312], synthetic [313], or immune [314] libraries, which contain the repertoire of human V genes. These libraries can be expanded utilizing display systems, such as yeast [315], ribosome [316], cDNA [317], and phage display [318]. Finally, the recovered sequence, that belongs to the most optimal monoclonal antibody can be cloned into the commercial plasmid that contains the desired Fc fragment. This cloned antibody can then undergo subsequent recombinant production, characterization, and validation.

In conclusion, this study provided a comprehensive description of significant cancer biomarkers and mechanisms of action of antibodies currently utilized in therapy. It offered an overview aimed at guiding future research toward potential approaches for the development of more accessible antibodies applied in immunotherapy, as well as various biotechnological applications.

## Figures and Tables

**Figure 1 biomedicines-11-02086-f001:**
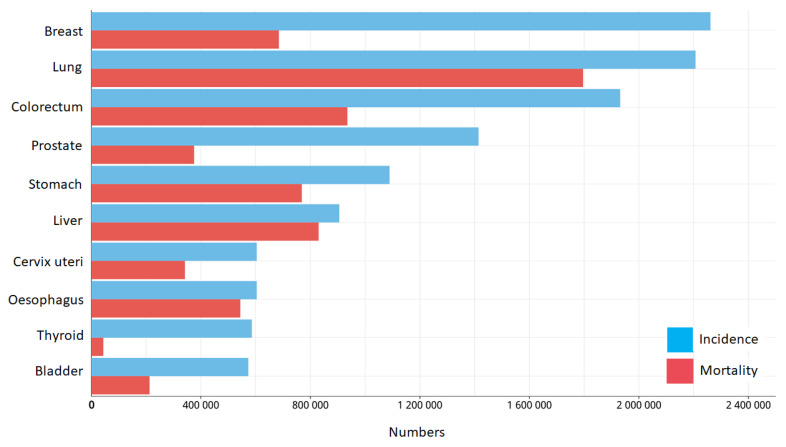
Estimated global number of new cancer cases in 2020, for both sexes and all age groups. Reproduced from Global Cancer Observatory (http://gco.iarc.fr/ (accessed on 1 July 2023)); data source: GLOBOCAN 2020, International Agency for Research on Cancer 2022.

**Figure 2 biomedicines-11-02086-f002:**
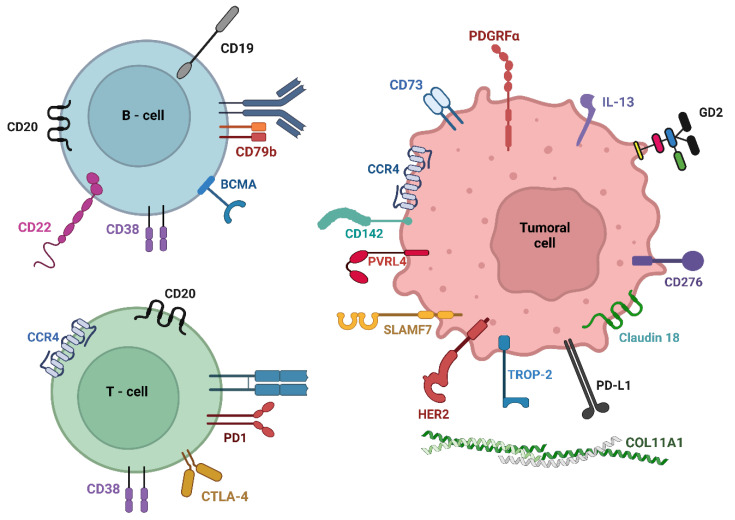
Tumoral antigens and therapeutic targets currently utilized in cancer immunotherapy are primarily located in cells of the immune system and tumor cells.

**Table 1 biomedicines-11-02086-t001:** Monoclonal antibodies (mAbs) approved for the use in the actual immunotherapy against cancer in both the United States (US) and Europe.

mAbs	Antigen that Recognizes	Format	Cancer for It Was Approved	Mechanism of Action	Reference
Dostarlimab	PD1	IgG4-humanized	Endometrial Cancer	Produced from a mouse hybridoma that acts as a PD-1 blocker via steric impediment with PD-L1 and PD-L2, thereby normalizing the immune response.	[279]
Cemiplimab	PD1	IgG4-humanized	Cutaneous squamous cell carcinoma	This mAb binds to PD1 on T-cells, blocking the interaction with PDL-1 and PDL-2 ligands and activating the immune response.	[280]
Durvalumab	PD-L1	IgG1-human	Bladder cancer	This is a human mAb with high affinity by PD-L1 and CD80.	[281]
Avelumab	PD-L1	IgG1-human	Merkel cell carcinoma	Regulates cytotoxicity mediated by antibody-dependent cells, due to the fact that it presents a native Fc region.	[282,283]
Atezolizumab	PD-L1	IgG1-humanized	Bladder cancer	Inhibits the interaction between PD-1 and B7.1, restoring the antitumor function of T cells.	[284]
Retifanlimab	PD-L1	IgG4-humanized	Merkel cell carcinoma	Blocks PD-1 interaction with its PD-L1 and PD-L2 ligands.	[30]
Mosunetuzumab	CD20-CD3	IgG1-humanized bispecific	Follicular lymphoma	Simultaneously binds to CD20 on malignant B cells and CD3 on T cells, causing T-cell activation and B-cell elimination.	[39,285]
Epcoritamab	CD20-CD3	IgG1-humanized bispecific	Diffuse large B-cell lymphoma	Induces T cells to kill CD20^+^ tumor cells through a unique mechanism of action (MOA).	[40,286]
Margetuximab	HER2	IgG1-chimeric	HER2^+^ breast cancer	Designed against HER2 to decrease binding to the inhibitory receptor Fcγ IIB (CD32B) and to increase binding to the receptor Fcγ activation IIA (CD16A).	[287]
Fam-Trastuzumab Deruxtecan	HER2	IgG1-humanized antibody–drug conjugate	HER2^+^ breast cancer	Designed against HER2 and conjugated to a cytotoxic topoisomerase 1 inhibitor	[80,288]
Loncastuximab Tesirine	CD19	IgG1-humanized antibody–drug conjugate	Diffuse large B-cell lymphoma	Designed to target CD19 and conjugated to a pyrrolobenzodiazepine DNA-alkylating warhead. It produces DNA interstrand crosslinks with high efficiency, leading to triggering of cell death.	[289,290]
Tafasitamab	CD19	IgG1-humanized	Diffuse large B-cell lymphoma	Produces antibody-dependent cellular cytotoxicity and antibody-dependent cell-mediated phagocytosis.	[291,292]
Naxitamab	GD2	IgG1-humanized	Neuroblastoma	Induces complement-dependent and cell-mediated antibody-dependent cytotoxicity	[293]
Teclistamab	BCMA-CD3	IgG4-humanized bispecific	Multiple myeloma	Redirects CD3-positive cells to BCMA-expressing tumor cells, inducing cytotoxicity and apoptosis.	[294]
Belantamab mafodotin	BCMA	IgG1-humanized antibody–drug conjugate	Multiple myeloma	Composed of an antibody that targets B-cell maturation antigen (BCMA), conjugated to the microtubule inhibitor monomethyl auristatin F (MMAF). The other part of the antibody binds to BCMA on the surface of the tumor cell, delivering the cytotoxic microtubule inhibitor MMAF to the therapeutic target.	[295]
Sacituzumab govitecan	TROP-2	IgG1-humanized antibody–drug conjugate	Triple-negative breast cancer	This mAb acts against TROP-2 conjugate with the active metabolite of irinotecan and topoisomerase 1 inhibitor.	[296,297]
Isatuximab	CD38	IgG1-chimeric	Multiple myeloma	By binding to CD38, this mAb causes apoptosis via multiple mechanisms such as antibody-dependent cellular phagocytosis, complement-dependent cytotoxicity, and effects that depend on the Fc region.	[298,299,300]
Enfortumab vedotin	Nectin-4	IgG1-human antibody–drug conjugate	Urothelial cancer	Works by releasing monomethyl auristatin E (MMAE) into cells that express nectin-4, causing apoptosis.	[301]
Polatuzumab vedotin	CD79b	IgG1-humanized antibody–drug conjugate	Diffuse large B-cell lymphoma	Works by binding to CD79b upon entering the cell, releasing MMAE, inhibiting cell division, and inducing apoptosis.	[124]
Moxetumomab pasudotox	CD22	IgG1-murine dsFv	Hairy cell leukemia	This mAb is conjugated with a toxic fragment of A exotoxin from Pseudomonas aeruginosa, which is internalized, resulting in apoptotic cell death.	[137,302]
Inotuzumab ozogamicin	CD22	IgG4-humanized antibody–drug conjugate	Hematological malignancy	By binding to CD22, the cytotoxic derivative of calicheamicin enters the cell, causing apoptosis.	[303]
Mogamulizumab	CCR4	IgG1-humanized	Cutaneous T-cell lymphoma	Has a defucosylated Fc region that enhances its antibody-dependent cellular cytotoxicity.	[304,305]
Olaratumab	PDGRFα	IgG1-human	Soft-tissue sarcoma	Blocks PDGF ligand biding and inhibits PDGFRα.	[306]
Elotuzumab	SLAMF7	IgG1-humanized	Multiple myeloma	Induces antibody-dependent cellular cytotoxicity and the activation of natural killer cells.	[307]
Tisotumab vedotin	CD142	IgG1-human antibody–drug conjugate	Cervical cancer	This is a human mAb conjugated with an antimitotic monomethyl auristatin E, which inhibits cell division by blocking polymerization of tubulin.	[183]
Tremelimumab	CTLA-4	IgG2A-human	Liver cancer	Blocks the union of B7-1 and B7-2 to CTLA4.	[308,309]

## Data Availability

Data Sharing not applicable.

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
