# Peer review of "Therapeutic Targets of Monoclonal Antibodies Used in the Treatment of Cancer: Current and Emerging"

_biomedicines, 2023, doi:10.3390/biomedicines11072086_

Round 1
Reviewer 1 Report
Dear Authors,
Overall, the manuscript addresses an important topic in mAb immunotherapy targets. However, I have several concerns that I believe need to be addressed before considering publication in this journal. Therefore, I recommend major revisions to improve the manuscript's quality and scientific rigor.
Major points;
1. While the authors extensively explain the structures and functions of the target molecules, there is a significant gap in the manuscript regarding the mechanisms by which each monoclonal antibody (mAb) treats cancers. Antibody therapies can work through various mechanisms, such as blocking ligand binding, inhibiting signal transduction, or inducing antibody-dependent cellular cytotoxicity (ADCC). Including information on the specific mechanisms of action for each mAb discussed would greatly enhance the manuscript's quality and provide a more comprehensive understanding of the topic.
2. The perspective section of the manuscript is challenging to understand and lacks a clear conclusion. For example, the statement on lines 613-614, "strategies that lead to the more accessible production of antibodies" and "One of these strategies consists—" do not connect well. The first part discusses the inexpensive supply of mAbs, while the latter part seems to address research tools for identifying effective mAbs. Clarifying the connection between these two statements and providing a clear conclusion in the perspective section would improve the manuscript's coherence and overall message.
Minor points;
· Section 2.9; Please cite the example for the currently available Nectin4 clinical mAb?
· line 98: the sentences before and after “however” do not connect.
· Line 103: therapeutic agents > therapeutic targets
· Line 117: the sentence “transcripts encoding truncated CD20 have been identified in malignant B-cells in order to avoid recognition by mAbs” is misleading. An expression like “which evade recognition by—” would be better.
· Please cite the references in English.
There are grammatical errors and somewhat awkward wording here and there. The authors may benefit from English editing services.
Author Response
Dear reviewer, below you will find an attached file with the responses to your comments. We hope you like them.
Regards

Reviewer 2 Report
This review describes the therapeutics targets actually used for Ab-based, cancer immunotherapy, and those that are emerging in the next future. The review holds significance because of the growing application of antibodies in cancer treatment.
My comments and suggestions.
The abstract section is uninformative. It's suitable to categorize the target molecules into groups in this section and specify the potential for their expansion.
The MS is extremely poorly structured and heavy on information, so it needs to be completely restructured. Data should be grouped into sections based on the mechanism of action of the antibodies (immunostimulation, cytotoxicity ....) or based on their practical application (lung cancer, breast caster...).
The MS is somewhat overloaded with secondary data about the target molecules. The MS would have been read with great interest if the it had included more information on the biological and clinical aspects of the use of therapeutic antibodies.
The perspective section does not outline in any way the clinical prospects for the use of therapeutic antibodies specific to novel target molecules.
Author Response
Dear reviewer, below you will find an attached file with the responses to your comments. We hope you like them.
Regards.

Round 2
Reviewer 1 Report
Dear authors,
Thank you for taking my comments into account. I found the updated manuscript is significantly more informative and meaningful. I believe this review would help researchers worldwide understand the importance of mAb medicine.
I noticed a possible typos; can line 692 “selective mating” be “selective binding”? But otherwise, I see no objection to accepting this manuscript in its current form.
Reviewer 2 Report
The authors have made great strides in improving the MS. All remaining remarks are non-essential.